# Compositional Multi-Object Reinforcement Learning with Linear Relation Networks

**Davide Mambelli** [*1,2] **Frederik Träuble** [*2]

**Stefan Bauer** [†3] **Bernhard Schölkopf** [†2] **Francesco Locatello** [†4]

[1]ETH Zürich, Switzerland, [2]Max Planck Institute for Intelligent Systems, Tübingen, Germany
[3]KTH, Stockholm, Sweden, [4]Amazon Web Services

## Abstract

Although reinforcement learning has seen remarkable progress over the last years, solving robust dexterous object-manipulation tasks in multi-object settings remains a challenge. In this paper, we focus on models that can learn manipulation tasks in fixed multi-object settings *and* extrapolate this skill zero-shot without any drop in performance when the number of objects changes. We consider the generic task of bringing a specific cube out of a set to a goal position. We find that previous approaches, which primarily leverage attention and graph neural network-based architectures, do not generalize their skills when the number of input objects changes while scaling as $K^2$. We propose an alternative plug-and-play module based on relational inductive biases to overcome these limitations. Besides exceeding performances in their training environment, we show that our approach, which scales linearly in $K$, allows agents to extrapolate and generalize zero-shot to any new object number.

## 1 Introduction

Deep reinforcement learning (RL) has witnessed remarkable progress over the last years, particularly in domains such as video games or other synthetic toy settings (Mnih et al., 2015; Silver et al., 2016; Vinyals et al., 2019). On the other hand, applying deep RL on real-world grounded robotic setup such as learning seemingly simple dexterous manipulation tasks in multi-object settings is still confronted with many fundamental limitations being the focus of many recent works (Duan et al., 2017; Janner et al., 2019; Deisenroth et al., 2011; Kroemer et al., 2018; Andrychowicz et al., 2020; Rajeswaran et al., 2018; Lee et al., 2021; Funk et al., 2022).

The reinforcement learning problem in robotics setups is much more challenging (Dulac-Arnold et al., 2021). Compared to discrete toy environments, state and action spaces are continuous, and solving tasks typically requires long-horizon time spans, requiring the agent to apply very long sequences of precise low-level control actions. Accordingly, exploration under easy-to-define sparse rewards becomes nearly impossible without horrendous amounts of data. This is usually impossible in the real world but has been likewise hard to achieve with computationally demanding realistic physics simulators. One promising path is to constrain the possible solution space of a learning agent by encoding suitable inductive biases in their neural architecture (Geman et al., 1992). Choosing good inductive biases that leverage the underlying structure of the problem setting can help to learn solutions that facilitate desired generalization capabilities (Mitchell, 1980; Baxter, 2000; Hessel et al., 2019). In the case of robotics, environments of multi-object manipulation tasks naturally accommodate a compositional description of their current state in terms of symbol-like entities (such as physical objects, robot parts, etc.). These representations can be directly obtained in simulator settings and ultimately hoped to be inferred robustly from learned object perception modules (Greff et al., 2019; Kipf et al., 2021; Locatello et al., 2020). While such a compositional understanding is in principle considered crucial for any systematic generalization ability (Greff et al., 2020; Spelke, 1990; Battaglia et al., 2018; Garnelo et al., 2016) it remains an open question

---

[*]Equal contribution. Correspondence: `d.mambelli@tudelft.nl` and `frederik.traeuble@tuebingen.mpg.de`.
[†]Equal Advising. Work done outside of Amazon.

how to design an agent that can process this type of input data to leverage this promise. Consider transporting a cube to a specified target location and assume the agent has mastered solving this task with three available cubes (see Figure 1 left). We refer to cubes different from the one to transport as distractors. This problem setting forms the basis of our work. If we now introduced two additional cubes, the solution to solve this task would remain the same, but the input distribution changed. So why not just train with much more cubes and use a sufficiently large input state? First, the learning problem requires exponentially more data when solving such a task with more available cubes (see Figure 10, red line), making it infeasible already for only half a dozen cubes.

Second, we do not want to put any constraint on the possible number of cubes at test time but simply might not have this flexibility at train time. Therefore, we demand to learn challenging manipulation tasks in fixed multi-cube settings and extrapolate this skill zero-shot without any drop in performance when the number of cubes changes. Achieving this objective will be the primary goal of this work. An essential prerequisite for achieving this is to endow the agents with the ability to process a variable number of input objects by choosing a suitable inductive bias. One potential model class for these object encoding modules are graph neural networks (GNNs), and a vast line of previous approaches builds upon the subclass of attentional GNNs to achieve this (Zambaldi et al., 2018; Li et al., 2020; Wilson and Hermans, 2020; Zadaianchuk et al., 2021). As a first contribution, we will show in Section 4 that utilizing such attention-based GNN approaches will not achieve the desiderata described above on our problem setting. Attentional approaches typically fail to extrapolate any learned behavior to a changing number of objects and require fine-tuning, resulting in catastrophic forgetting of previous skills. Lastly, using attention scales quadratically in the number of input objects which becomes impractical for training and inference with many objects.

**Main contributions.** In this work, we present support to build agents upon relational reasoning inductive biases. Specifically, we propose an object encoder module based on a linear relation neural network (LRN)

Train agent on manipulation task with **N = 2 distractors**

Zero-shot generalization to any **N ≠ 2 distractors**

T = 0

T = 0

T = End

T = End

Figure 1: Consider the following **problem setting**: There are 3 identical cubes in the arena. During an episode, a random cube is selected (for simplicity, color-coded in blue in the figure) to be transported to an episode-specific location (for simplicity, color-coded as a green cube in the figure). The remaining two cubes are distractors (color-coded in red). Can we train RL agents that learn such manipulation tasks with a fixed multi-object setting – training with only two distractors – and extrapolate this skill zero-shot when the number of distractors changes?

that is able to learn challenging manipulation tasks in fixed multi-object settings. We show that our module consistently outperforms an attention-based baseline while scaling only linear in the number of objects. Most importantly, we show that our agents can extrapolate and therefore generalize zero-shot without any drop in performance when the number of objects changes. We present supporting evidence for this compositional generalization capability and necessary requirements for multi-object manipulation in two challenging task environments.

## 2 PROBLEM SETTING AND RELATED WORK

Consider the following problem setting that we summarize in Figure 1: There are 3 identical cubes in an arena. A random cube is selected to be transported to an episode-specific location during each episode. The remaining 2 cubes are then distractors for this episode. This work aims to answer the following question: Can we train RL agents that learn such manipulation tasks from a fixed multi-cube setting and extrapolate this skill zero-shot when the number of distractors changes?

This work considers tasks where the goal is always to transport one of the $N_o$ movable objects, i.e. cubes, to a target location, and all other $N_o - 1$ cubes act as distractors. Therefore, given their straightforward relation, we refer to each environment either using the total number of cubes or the number of distractors, the latter called $N$. Specifically, we define the objective of any learning agent as follows: First, the agent needs to learn to solve the task at train time under a *constant* and a minimal number of manipulable objects $N_o^{train}$. Second, the agent needs to be able to *zero-shot* extrapolate solving the equivalent task under $N_o^{test} \neq N_o^{train}$ objects in the scene. Therefore, the agent needs to learn an invariant behavior with respect to the episode-specific distractors. Note that our task formulation can be seen as generic subtasks of more general manipulation tasks. If we can

solve this task, *any* more complex multi-object goal can be solved by specifying one object-goal pose after another. Crucially, we want to learn such tasks with a constant number of available objects because varying the number of objects during training might limit us in the real world. We ultimately need adapting agents that can generalize when additional objects are added or removed from the environment scene. We begin by choosing train environment with at least three movable objects to let agents also experience potentially important three-body interactions during training.

**Background on compositional multi-object manipulation tasks.** Given the compositional environment structure of such real-world grounded manipulation tasks, we are concerned with input states $s_t \in S$ that comprise a set of objects describing individual rigid objects in the scene such as movable cubes $\{s_1^o, s_2^o, ..., s_{N_o}^o\}$ or the individual parts of a robot such as each finger in our setup $\{s_1^r, s_2^r, ..., s_{N_r}^r\}$. To facilitate a goal-conditioned formulation, we similarly feed additional environment-specific information in the form of a goal object $s^g$. Concerning the objective of our tasks, the goal information can be provided via the object state $s^g$ that describes the target pose of this cube. To identify the object to be transported, we can simply add a unique identifier feature to each object state and specify the object to be moved through the same identifier in the goal object. Translated to a robotic RL setting, the above state-space factorization requires a policy that is parameterized by a function $a_t = \pi(\{s_1^o, ..., s_{N_o}^o, s_1^r, ..., s_{N_r}^r, s^g\})$. Our demand is that this policy can deal with a variable number $N_o$ of input objects independent of their order. In other words, the policy needs to be permutation-invariant.

**Domain specific related works.** Flavours of this problem have a long history and are typically summarized as relational reinforcement learning, which tries to approach this by various suitable inductive biases that enable reasoning upon such object-oriented structured representations (Džeroski et al., 1998; Džeroski et al., 2001; Kaelbling et al., 2001; Van Otterlo, 2002). A popular class of neural architecture that offers these properties are various sub-types of graph neural networks (GNNs) with attention-based approaches or interaction networks being one special case (Bronstein et al., 2021; Veličković et al., 2018; Vaswani et al., 2017; Battaglia et al., 2016; Santoro et al., 2017; Kipf and Welling, 2017). Zambaldi et al. (2018) was one of the first to leverage the self-attention approach in the context of deep reinforcement learning for toy and game environments from a constant number of features extracted from a CNN. The ATTN baseline in our experiments directly resembles their object-processing module, but we will also evaluate this approach when the number of input objects changes. Interestingly, most of the previous work for learning similar manipulation tasks in multi-object settings uses such object-encoding modules based on multi-head dot product attention (MHDPA) layers to process the object-based input representations Li et al. (2020); Zadaianchuk et al. (2021); Wilson and Hermans (2020). In summary, previous approaches suffer from various fundamental limitations regarding our desiderata. First, the compute of MHDPA-based architectures scales quadratically with the number of objects, which is especially limiting for training and inference with $N_o \gg 1$. Second, to achieve extrapolation for different object numbers, the training domain must cover a variable and typically large number of objects. Third, the performance at test time typically decreases for a new number of objects, and fine-tuning is required to recover the original performance.

**Related work on relation networks.** Our method is motivated by means of an alternative perspective on this problem via relation networks (Raposo et al., 2017; Santoro et al., 2017). Relation networks process input sets by computing pairwise relations for every object-object pair, followed by a permutation invariant aggregation function, such as the sum, mean, or max. Relation networks generally scale quadratically in the number of input objects (Santoro et al., 2017). They are predominantly used for various non-RL applications such as visual and text-based question answering, few-shot image comparison, or object detection (Santoro et al., 2017; Sung et al., 2018; Hu et al., 2018). Importantly, we are not aware of RL applications approaching this problem utilizing relation networks as relational inductive biases. In addition, standard relation networks typically operate on a fixed number of input objects and suffer from the same scalability issues as attention.

## 3 AGENTS WITH OBJECT REASONING MODULES

This section describes the linear relation network (LRN) that we propose as an alternative object reasoning module for learning compositional multi-object manipulation skills. See Figure 2 for a summary of the overall architecture, and the agent model is described in Appendix A.4.

**Linear Relation Network Reasoning Module.** Similar to the original Relation Network from Santoro et al. (2017) used for visual question answering tasks, our module builds upon the idea

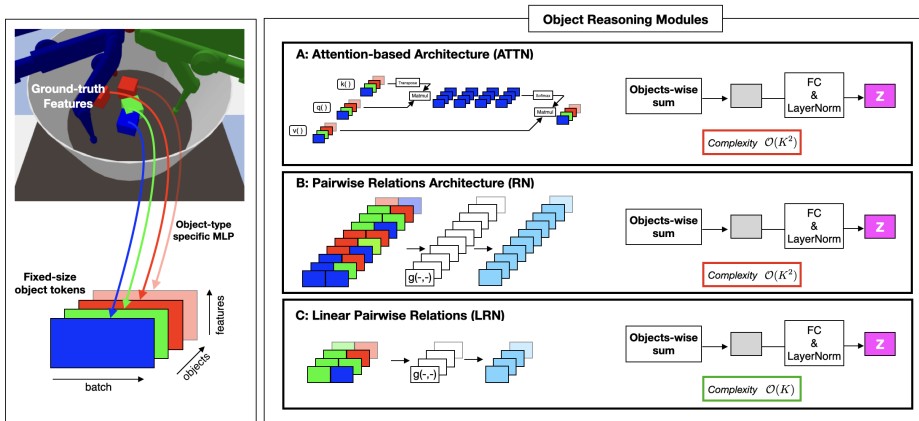

Figure 2: Summary of object reasoning modules for learning challenging manipulation skills studied in this work. Left: In a first step, ground-truth feature states are encoded into fixed-size object tokens via object-type specific small MLPs. Right: An object-encoder module processes this token set and outputs a fixed-size representation (highlighted in pink). This representation forms the input of any downstream predictor required to learn the optimal policy. Besides an attention based and vanilla relation network object encoder module (Box A and B), we propose to use our Linear Relation Network architecture (LRN) which only uses relations involving the goal object, thereby scaling linearly in the number of input tokens (Box C).

of reasoning about the future actions from learned relations between the given objects. It seems reasonable that in the context of goal-conditioned RL, the next optimal action is primarily governed by the relations of the available objects with the particular goal of interest. In contrast, most other relations might be only important in a sparse and context-dependent way. We, therefore, argue to focus on these relations, and we propose the following adjusted composite representation function

$$\mathbf{z} = \text{LayerNorm}\left(\mathbf{f}_\phi\left(\sum_i^K \mathbf{g}_\theta(\mathbf{o}_i, \mathbf{o}_g)\right)\right), \tag{1}$$

where the $\mathbf{o}_i \in \mathbb{R}^d$ are common-size embeddings of the K input objects. The objects are the scene's moveable elements in our setting, which means cubes and the three robotic finger states. The goal information of the task is being embedded into a separate object $\mathbf{o}_g \in \mathbb{R}^d$. First, the shared learnable function $\mathbf{g}_\theta$ computes the K embeddings $\mathbf{r}_i \in \mathbb{R}^d$ for each object-goal relation. For our experiment $\mathbf{g}_\theta$ is a 4-layer MLP. Next, a dimension-wise sum operation over the K relations is applied, which serves as a permutation-invariant aggregation function. The final representation $\mathbf{z} \in \mathbb{R}^{d_z}$ is obtained after processing this intermediate representation by another learnable function $\mathbf{f}_\phi$ followed by a layer normalization operation (Ba et al., 2016). The function $\mathbf{f}_\phi$ can serve the purpose of learning further necessary relations from the K object-goal relations. In our experiments, we will use a simple linear transformation for $\mathbf{f}_\phi$. The layer normalization ensures that the final representation $\mathbf{z}$ remains in distribution for a changing number of input objects. The overall complexity of this goal-conditioned linear relation network reasoning module is linear in K, i.e. $\mathcal{O}(K)$. To assess the impact of inferring the K object-goal relations only, we will also implement an equivalent variant that considers *all* possible $K^2$ relations, similar to the naive implementation of relation networks ('RN').

## 4 EXPERIMENTS

We now aim to evaluate the previously described and proposed architectures for learning challenging manipulation tasks in fixed multi-object settings *and* their ability to extrapolate this skill zero-shot when the number of objects changes. For an outline of the experimental setup refer to Appendix A.7. Here, we will discuss the learning behaviour in terms of sample efficiency and performance of our proposed modules against the common attention-based approach from previous works in Section 4.1. We close this section by demonstrating the models' zero-shot extrapolation skills in Section 4.2.

### 4.1 LEARNING TO MANIPULATE OBJECTS FROM FIXED MULTI-OBJECT TRAIN DOMAINS.

Quantitative results for learning to manipulate objects from the 3 cubes train domain are summarized in Figure 3 for PushingCube in the top panel and LiftingCube in the lower panel. We refer to the project-site (sites.google.com/view/compositional-rrl/) for additional videos. Across tasks, we can

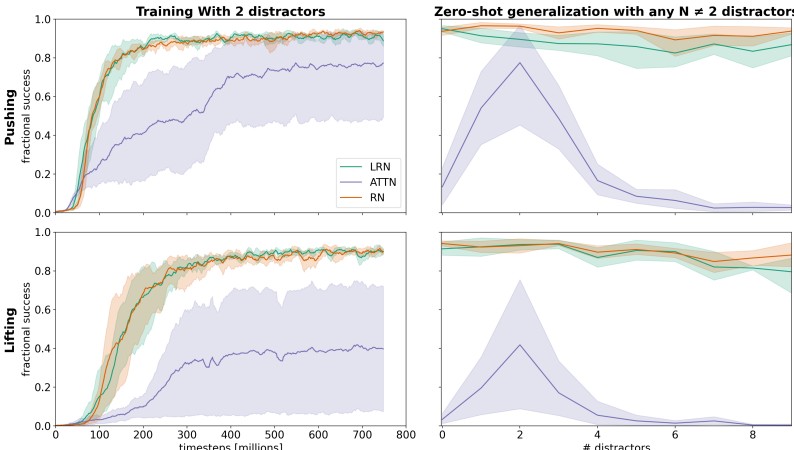

Figure 3: **Training curves and zero-shot generalization from fixed multi-object train domains.** Left: Training curves for PushingCube (top) and LiftingCube (bottom) tasks in three-object environments (5 random seeds per model). Our RN and LRN modules perform favourably against attention-based implementation in terms of final success, sample efficiency, and variance across different seeds. Right: Evaluating the corresponding zero-shot generalization on equivalent N-object environments for both tasks. Our LRN and RN modules show a strong extrapolation ability of their learned manipulation skills to environments with many more objects.

see that our proposed backbone performs very favourably over the attention-based ATTN baseline. Specifically, both relation network variants – RN and LRN – reliably succeed in learning to solve the tasks almost perfectly with very high scores. On the other hand, ATTN exhibits much higher variance across different seeds and takes longer to train. ATTN is clearly inferior in terms of final performances. Interestingly, RN and LRN reveal very similar robust learning behaviour in terms of training time steps. **Summary.** The relation network reasoning modules and our LRN model represent a promising alternative over attention-based architectures commonly used in prior works for learning to manipulate an object on a multi-object train domain. Our module outperforms these approaches in final success, sample efficiency, and variance across different seeds

### 4.2    ZERO-SHOT GENERALIZATION TO N-OBJECT ENVIRONMENTS.

We now evaluate the trained models' generalization ability when the number of objects is different from training. Our main result to assess this question is summarized in Figure 3 (right). Remember that all agents have been trained for pushing and lifting an object in 3 cubes environments, synonymous with 2 distractors. Testing these agents' generalization in N-object environments with $N_o \neq 3$ reveals a clear separation between the relation network modules and the attention-based architecture. Our RN and LRN modules can maintain strong train-time performance for any tested number of objects. On the other hand, agents with the ATTN module cannot generalize to a changing number of objects and experience a strong and consistent fading ability to solve the task the more the number of objects deviates from the training environment. ATTN-based agents fail to generalize under our test tasks completely when adding more than 2 additional objects. This result contrasts with the RN and LRN module-based agents, who can still reliably solve these tasks. Because we see no relevant deteriorating trend, we suspect that this generalization capability could potentially even hold beyond the 9-distractor environments tested here. Interestingly, the same trends can be seen for environments with less than 3 cubes. Even though it seems intuitive that an agent should be able to solve pushing or lifting an object when no further distractors are in the scene, the ATTN module seems to overfit on the specific 2-distractor input train distribution strongly and cannot solve these tasks without exactly two distractors. On the other hand, RN and LRN module-based agents can robustly maintain their respective train-time performances, presumably due to their relation-based processing of the input objects. **Summary.** Agents based on the relation network modules show a strong extrapolation ability of their learned manipulation skills to environments with many more objects. In contrast, we do not observe such zero-shot generalization with commonly employed attention-based architectures, which experience a substantial decrease in performance when deployed out-of-distribution.

## 5 CONCLUSION

In this work, we presented a new object encoder module based on relation networks, enabling reinforcement learning agents to learn challenging compositional manipulation tasks in multi-object robotics environments. We extensively studied a newly proposed linear relation network module on the generic task of transporting one particular cube out of N cubes to any specified goal position. While previous approaches mostly build upon quadratically scaling attention-based architectures to process the set of input objects, our approach scales only linear with the number of objects. At the same time, our method outperforms these previous approaches at train time. Most importantly, we can show that our agents can extrapolate and generalize zero-shot without any drop in performance when the number of objects changes which is a major limitation of most previous approaches.

## ACKNOWLEDGEMENTS

We thank the International Max Planck Research School for Intelligent Systems (IMPRS-IS) for supporting FT.

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

| | type | size | position | orientation | linear velocity | angular velocity | identifier | end-effector position |
|---|---|---|---|---|---|---|---|---|
| Cube | 1 | 3 | 3 | 4 | 3 | 3 | 1 | - |
| Goal | 1 | 3 | 3 | 4 | - | - | 1 | - |
| Finger | - | - | 3 | - | 3 | - | 1 | 3 |

Table 1: Number or variables used to describe each element of the state for each object type

## A  IMPLEMENTATION DETAILS

### A.1  REWARD DESIGN

We used the following same reward for both tasks:

$$
\begin{aligned}
R_t = {} & \alpha_1 \rho_t \\
& + \alpha_2 \left[ d(o_t, g_t) - d(o_{t-1}, g_{t-1}) \right] \\
& + \alpha_3 (1 - \rho_t)^\beta \log \left[ d(o_t, o_{t-1}) + 1e - 5 \right]
\end{aligned}
\tag{2}
$$

where $t$ denotes the time step, $\rho_t \in [0, 1]$ is the fractional overlap with the goal cube at time $t$, $e_t \in \mathbf{R}^3$ is the end-effector position, $o_t \in \mathbf{R}^3$ the cube position of the object that is supposed to be used for the task, $g_t \in \mathbf{R}^3$ the goal position, and $d(\cdot, \cdot)$ denotes the Euclidean distance. Across all experiments we set $\boldsymbol{\alpha} = [100, 250, 10]$ and $\beta = 0.05$.

### A.2  TRAINING SCHEME FOR AGENTS USING PPO

We use the distributed PPO implementation from `rlpyt` (Stooke and Abbeel, 2019) with discount factor 0.98, entropy loss coefficient 0.01, learning rate 0.00025, value loss coefficient 0.5, gradient clipping norm 0.5, 40 minibatches, gae lambda 0.95, clip ratio 0.5 and 4 epochs. We applied a linear learning rate scheduling and Adam as optimizer (Kingma and Ba, 2015).

### A.3  SIMULATOR INPUT FEATURES

The ground truth representations for each object type are described in Table 1. Cubes are described using their position and orientation and each corresponding velocity. In addition, every cube is marked with a unique identifier which makes the set of cubes heterogeneous. Similarly, the goal is specified via its cartesian position, orientation, and an identifier variable used to specify which cube has to be moved. Both cubes and the goal include features describing their shape and size; however, we always use a cube shape and fixed size in this work. Each finger of the robot has 3 degrees of freedom, and it is described by a 9-dimensional vector composed by its joint positions, joint velocities, and end-effector position.

### A.4  AGENT MODEL

Our agents will build upon the following standard architecture scaffold. In a first step, ground-truth simulator state features represented by objects are encoded into fixed-size object tokens $\mathbf{o}_i \in \mathbb{R}^d$ via object-type specific small MLPs. In our case, these are three robotic states for each finger, goal objects, and physical objects for the cube. Next, an object-reasoning encoder module processes this token set and outputs a fixed-size representation $\mathbf{z}$ (highlighted in pink in Figure 2). Finally, this representation forms the input of any downstream predictor required to learn the optimal policy. Here, we will assume a small MLP to predict the action (policy); however, we can likewise use the representation to predict the value (value network) or Q-Value (when combined with an action) when using Q-Learning or actor-critic approaches. The primary focus of this work is on proposing a simple and flexible object-encoder module that overcomes the limitations of previous approaches.

### A.5  ATTENTION MODULE - BASELINE

The baseline we use in this work exploits a MHDPA module (Vaswani et al., 2017) operating on the set $\mathbf{o} \in \mathbb{R}^{d \times (K+1)}$ composed of common-size embedding of the K input objects $\mathbf{o}_i \in \mathbb{R}^d$, and the goal $\mathbf{o}_g \in \mathbb{R}^d$ with

$$
\mathbf{o}^{attn} = \mathrm{softmax}\left( \frac{q(\mathbf{o})k(\mathbf{o})^T}{\sqrt{d}} \right) v(\mathbf{o})
\tag{3}
$$

where $\mathbf{o}^{attn}$ is the set of $K + 1$ embeddings scaled from the MHDPA module. In this module the goal-conditioning element is obtained by dealing with the goal as an object. This set is then mapped

to the fixed-size embedding by an aggregation function and normalized to avoid changes in scale when the number of objects differ between training to testing.

$$\mathbf{z} = \text{LayerNorm}\left(\mathbf{f}_\phi\left(\sum_i^{K+1} \mathbf{o}_i^{attn}\right)\right) \tag{4}$$

This reasoning module resembles the one from Zambaldi et al. (2018), and it is used to introduce relational inductive biases into the learning process. The same self-attention layer is also used by Zadaianchuk et al. (2021) to build a goal-conditioned policy that is able to handle set-based input representations. Similarly, it also describes approaches that rely on Graph Attention Networks (GAT) (Veličković et al., 2018) to build policies while using the common assumption in multi-object manipulation of a fully connected graph (Li et al., 2020; Wilson and Hermans, 2020). Self-attention when computing the attention matrix takes into consideration each object-object pair, which makes it scale quadratically with the number of objects in the input set, as highlighted in Figure 2 (A).

### A.6 RELATION NETWORK MODULE

To have a further element of comparison, we introduced Relation Networks as relational reasoning module in the context of reinforcement learning.

$$\mathbf{z} = \text{LayerNorm}\left(\mathbf{f}_\phi\left(\sum_i^{K+1}\sum_j^{K+1} \mathbf{g}_\theta(\mathbf{o}_i, \mathbf{o}_j)\right)\right) \tag{5}$$

Relation Networks have been proposed in Santoro et al. (2017) and they compute $(K+1)^2$ embeddings $\mathbf{r}_i \in \mathbb{R}^d$, referred to as relations, using the shared function $\mathbf{g}_\theta$. Relations are aggregated using a sum and then normalized to obtain a fixed-size embedding used as input for the MLP policy head. Similarly to self-attention, Relation Networks need to consider every object-object interaction, making them scale quadratically in the number of objects. A key aspect of this architecture is that $\mathbf{g}_\theta$ always takes as input a pair of objects, independent of the total number of objects in the input set. This property makes the architecture robust to changes in the number of objects as long as the objects' statistics do not change.

### A.7 EXPERIMENTAL SETUP.

Both tasks are derived from the simulated robotic manipulation setup and tasks proposed in Causal-World (Ahmed* et al., 2021) which builds upon the robotic TriFinger design (Wüthrich et al., 2020; Bauer et al., 2021). Compared to previous work on this setup, we attempt to solve tasks in scenes that contain more than 1 cube (Dittadi et al., 2021; Ahmed* et al., 2021; Allshire et al., 2021). Specifically, the goal is to move any of the 3 available cubes to a freely specifiable target position. We cover a simpler sub-task where the target position is constrained to the floor and the more general task where the target position is also allowed to be above the ground. Such a lifting task involves learning to hold the cube stable in the air at a variable height. We refer to these tasks as PushingCube and LiftingCube, respectively. As discussed in Section 2, we specify the object to be rearranged through a matching object identifier in one of the goal objects' features. For both tasks, we use a generic reward function designed to work for any single-goal task in this environment. The reward is composed of three terms: (1) a reward term that directly measures the success at each time step, which is defined as the volumetric overlap of the cube with the goal cube; (2) a second reward term that encourages moving the target object closer to the goal; (3) a third curiosity-inspired reward term for efficient exploration that gives a positive reward signal for moving the target object when being further away from the goal. See Appendix C for an analysis regarding the role of each term. Across all tasks, we will report the factional success as the universal evaluation metric at the last timestep of an episode. Tasks can be visually considered solved with a score around 80%, and we can therefore ensure an objective and interpretable performance measure (Ahmed* et al., 2021; Dittadi et al., 2021). All models are being trained under the described reward with the help of PPO (Schulman et al., 2017) on distributed workers with 5 random seeds each. Initial object poses and the goal pose are randomly sampled at the beginning of each episode, and the episodic task needs to be solved within 30 simulation seconds or 2500 frames (10 seconds per object). We compare the instantiation of our proposed RN and LRN network architectures against a representative attention-based baseline derived from (Zambaldi et al., 2018), referred to as ATTN. We ensured a fair comparison of this baseline through extensive hyperparameter optimization (see Appendix D.3). We refer to the supplement for a comprehensive account of further implementation details.

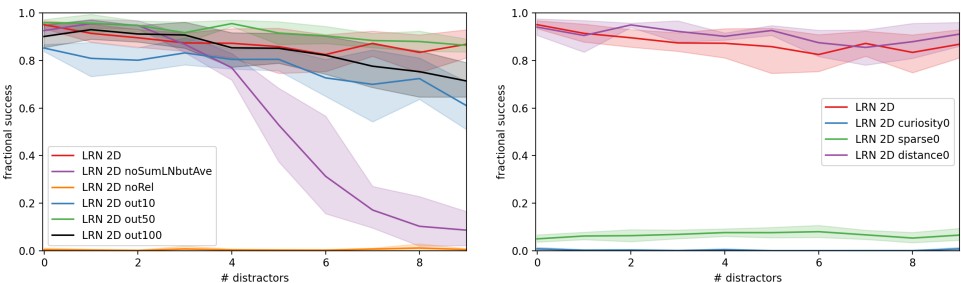

Figure 4: Zero-shot generalization of the linear relation network module on the PushingCube task concerning various architecture choices and the proposed reward structure. Left: We find that we cannot solve the task when dropping the object-goal relations (purple curve). We also observe that even though our agents can learn the task using an average instead of the LayerNorm, our model experiences a fading extrapolation performance (orange curve). Finally, we find a much weaker fading in generalization ability when providing too few or too many representational dimensions (green and red curves). Right: When ablating on the individual terms of our reward function, we first find that removing the distance term does not affect the overall performance (red curve). Whereas the sparse term only combined with the curiosity term is sufficient to solve the task successfully.

## B  DISCUSSION

**Role of changing the train domains.** In the previous section, we always trained under exactly 2 distractors. In particular, we believed that a 2-distractor environment also includes edge cases that the agent must face during training to solve the same task with a large number of distractors. An example is when the selected cube is stuck between 2 distractor cubes blocking its grasping. We found this to be a frequent scenario when the arena gets crowded. To validate this hypothesis, we repeated our experiments in environments with 3 distractors and additionally covered the limiting case of only one and no distractor at train time at all. Our results are presented in Figure 5, which shows how the zero-shot generalization under each architecture is affected by its training distribution. Concerning the ATTN module, we observe the effect of the train distribution on generalization capabilities to be generally limited. Performances always peak in the training environment and monotonically decrease for $N_o^{test} \neq N_o^{train}$. Whereas, a steady improvement can be observed for the RN module when changing the training environment up until 2 distractors. Note that the RN already achieves some level of generalization when training with a single distractor. It appears a phase change to robust performances happens when 2 distractors are used for training which yields strong generalization for any test environment. Finally, we can observe that our LRN module already achieves a strong generalization when trained with one distractor only, which outperforms both previous architectures. We therefore conclude that our assumption on the minimum number of distractors is not necessary for our LRN module.

**Assessing architectural choices of our LRN module.** We also tested the impact of various design choices for our proposed linear relation network module. Results are summarized in Figure 6 (left). Specifically, we focus on 3 questions: First, do we need relations at all? To answer, we repeated the LRN experiments but replaced the relations with equivalent non-relational single-object feedforward MLPs, i.e. $\mathbf{g}_\theta(\mathbf{o}_i)$. We found that the module cannot even learn the task on the training domain without relations. Second, we want to understand if we could use a simple average instead of the LayerNorm after summing over all relations? Our experiments show that even though we can learn the task on the train environments, the agents reveal deteriorating generalization the more objects we add to the task. We hypothesize this to be due to the changing variance of $\mathbf{z}$ which is kept constant when doing a layer normalization but not an average. Lastly, we analyzed the role of the representations' dimension, for which we used $d_z = 25$. When repeating the experiments with $d_z = 10$, the zero-shot generalization slowly decreases to about 60 % success with 9 objects. On the other hand, when allowing for $d_z = 50$ or even $d_z = 100$, the agent achieves slightly elevated train time performance with a slight drop of about 5 % and a more pronounced drop of about 20 % respectively when testing with up to 9 distractors. This result suggests that the representation

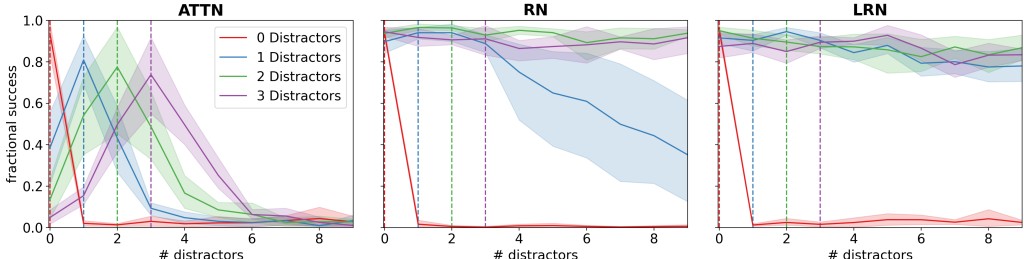

Figure 5: **Zero-shot generalization across different number of training distractors.** All three modules (left: ATTN; middle: RN; right: LRN) were retrained under 0, 1, 2, and 3 distractors with 5 seeds each. The ATTN baseline always peaks in performance for its train domain, quickly vanishing generalization to any new number of cubes. Crucially, our RN module can generalize to any number of objects in the scene if trained on at least 2 distractors (3 cubes in total). The LRN module achieves this with as little as 1 distractor at train time.

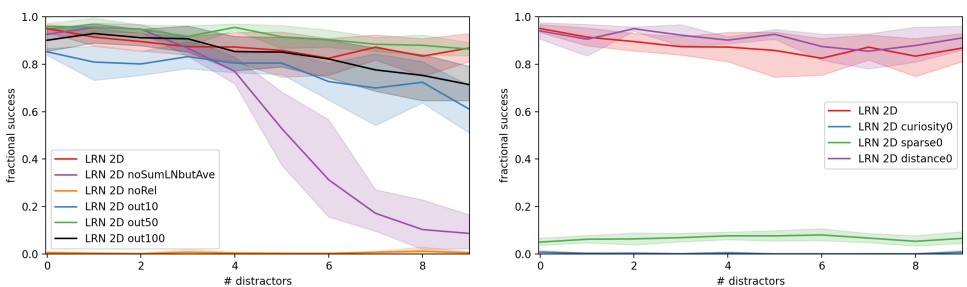

Figure 6: Zero-shot generalization of the linear relation network module on the PushingCube task concerning various architecture choices and the proposed reward structure. Left: We find that we cannot solve the task when dropping the object-goal relations (purple curve). We also observe that even though our agents can learn the task using an average instead of the LayerNorm, our model experiences a fading extrapolation performance (orange curve). Finally, we find a much weaker fading in generalization ability when providing too few or too many representational dimensions (green and red curves). Right: When ablating on the individual terms of our reward function, we first find that removing the distance term does not affect the overall performance (red curve). Whereas the sparse term only combined with the curiosity term is sufficient to solve the task successfully.

dimension is optimally being adjusted for the right capacity requirements of the task.

**Role of the reward function.** We finally wanted to understand the impact of each term on the reward function we proposed. Figure 6 (right) shows the generalization performance of corresponding agents trained with each variation of the reward function. We can distinguish reward terms based on their effect: sparse, and distance terms can be used to define the task; the curiosity term guides the exploration. The first notable observation is that removing the distance term does not affect the overall performance. Therefore, we can conclude that the sparse term is sufficient to solve the task successfully. On the other hand, removing the sparse term while keeping the distance term stops the agent from learning to solve the task, underlying its importance.

## C  DISCUSSION

**Role of changing the train domains.** In the previous section, we always trained under exactly 2 distractors. In particular, we believed that a 2-distractor environment also includes edge cases that the agent must face during training to solve the same task with a large number of distractors. An example is when the selected cube is stuck between 2 distractor cubes blocking its grasping. We found this to be a frequent scenario when the arena gets crowded. To validate this hypothesis, we repeated our

experiments in environments with 3 distractors and additionally covered the limiting case of only one and no distractor at train time at all. Our results are presented in Figure 5, which shows how the zero-shot generalization under each architecture is affected by its training distribution. Concerning the ATTN module, we observe the effect of the train distribution on generalization capabilities to be generally limited. Performances always peak in the training environment and monotonically decrease for $N_o^{test} \neq N_o^{train}$. Whereas, a steady improvement can be observed for the RN module when changing the training environment up until 2 distractors. Note that the RN already achieves some level of generalization when training with a single distractor. It appears a phase change to robust performances happens when 2 distractors are used for training which yields strong generalization for any test environment. Finally, we can observe that our LRN module already achieves a strong generalization when trained with one distractor only, which outperforms both previous architectures. We therefore conclude that our assumption on the minimum number of distractors is not necessary for our LRN module.

**Assessing architectural choices of our LRN module.** We also tested the impact of various design choices for our proposed linear relation network module. Results are summarized in Figure 6 (left). Specifically, we focus on 3 questions: First, do we need relations at all? To answer, we repeated the LRN experiments but replaced the relations with equivalent non-relational single-object feedforward MLPs, i.e. $\mathbf{g}_\theta(\mathbf{o}_i)$. We found that the module cannot even learn the task on the training domain without relations. Second, we want to understand if we could use a simple average instead of the LayerNorm after summing over all relations? Our experiments show that even though we can learn the task on the train environments, the agents reveal deteriorating generalization the more objects we add to the task. We hypothesize this to be due to the changing variance of $\mathbf{z}$ which is kept constant when doing a layer normalization but not an average. Lastly, we analyzed the role of the representations' dimension, for which we used $d_z = 25$. When repeating the experiments with $d_z = 10$, the zero-shot generalization slowly decreases to about 60 % success with 9 objects. On the other hand, when allowing for $d_z = 50$ or even $d_z = 100$, the agent achieves slightly elevated train time performance with a slight drop of about 5 % and a more pronounced drop of about 20 % respectively when testing with up to 9 distractors. This result suggests that the representation dimension is optimally being adjusted for the right capacity requirements of the task.

**Role of the reward function.** We finally wanted to understand the impact of each term on the reward function we proposed. Figure 6 (right) shows the generalization performance of corresponding agents trained with each variation of the reward function. We can distinguish reward terms based on their effect: sparse, and distance terms can be used to define the task; the curiosity term guides the exploration. The first notable observation is that removing the distance term does not affect the overall performance. Therefore, we can conclude that the sparse term is sufficient to solve the task successfully. On the other hand, removing the sparse term while keeping the distance term stops the agent from learning to solve the task, underlying its importance.

# D ADDITIONAL RESULTS

## D.1 DISTRIBUTION SHIFT FOR OOD TASKS

To better understand the difference between the architectures, we visualized the representations of each reasoning module with the help of lower-dimensional embeddings. Each model has been tested for 200 episodes on environments with the number of distractors ranging from 0 up to 9. The representations obtained as output have been mapped to a low-dimensional space using t-SNE (van der Maaten and Hinton, 2008). It is important to observe that these points represent the input of the MLP policy head; therefore, to have generalization capabilities, they should have similar statistics across different tasks. In Figure 7(a) we can observe the result obtained for the attention-based module when visualized in a 2-dimensional space. The embedding distribution changes with the task, which causes the MLP policy head to operate out-of-distribution justifying the drop in performances observed in Figure 3. In Figure 7(b) and Figure 7(c), respectively the RN and LRN output embeddings are visualized. For both, we can say that the approximated support of the distribution appears to be invariant with respect to the task.

## D.2 RELATIONS ANALYSIS

**Relations Sub-Spaces** We decided to visualize 2-dimensional embeddings computed using t-SNE color-coding different relations to understand whether relations with different objects are qualitatively different. We used relations generated from 200 episodes with a number of distractors varying from

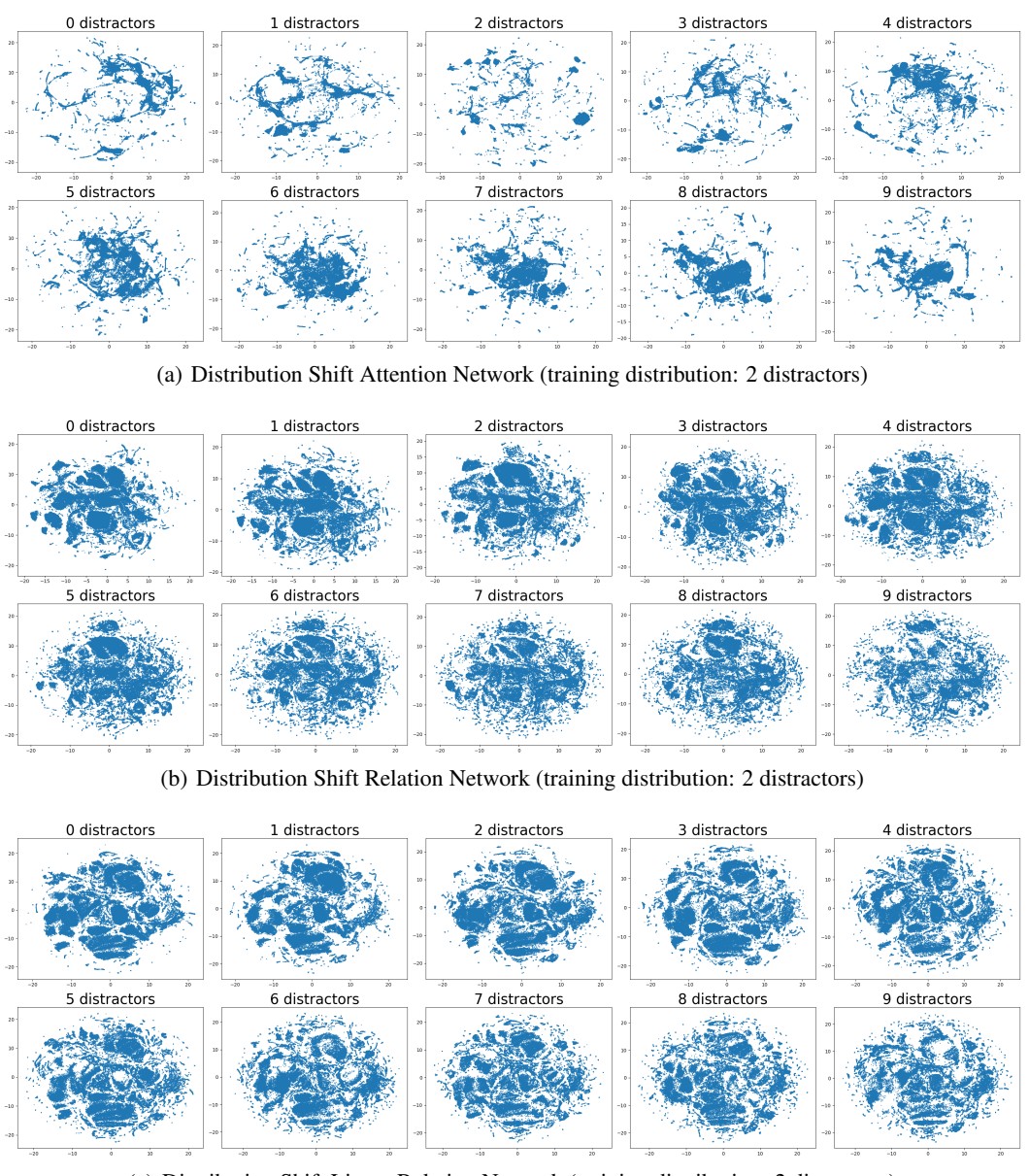

(a) Distribution Shift Attention Network (training distribution: 2 distractors)

(b) Distribution Shift Relation Network (training distribution: 2 distractors)

(c) Distribution Shift Linear Relation Network (training distribution: 2 distractors)

Figure 7: OOD Distribution shifts

3 to 6. The obtained plot is presented in Figure 8, where cube-goal relations are shown in green, distractors-goal relations are visualized in blue, and fingers relations are colored respectively in black, brown, and red. Different relations are mapped to different sub-spaces of the representation space, suggesting that invariance to the number of distractors might be obtained via a gating property learned by the architecture to filter out relations embedded into certain sub-spaces when computing the action.

**Relations over time** We also analyzed how the norm of each relation type changes over time while the agent is solving the task. In Figure 9 we show the result when testing on 5 distractors a LRN-based agent and normalizing each type of relation by the number of relations belonging to that type. Since we used the sum as the aggregation function, the norm can be associated with each relation's weight in the representation after the sum. Interestingly, the most important relations are those describing fingers and the target cube, whereas those linked to distractors have a much smaller norm. This qualitative result suggests that the cube-goal relation might be correlated with the cube-goal distance.

**Capturing higher-order relations** The Linear Relation Network does not compute most of the possible relations explicitly, which is why it scales linearly with the number of objects. A question that naturally arises is what happens when a particular relation that the Linear Relation Network does not compute is strictly necessary to solve the task. An example is provided by the pushing task, where the agent must reason about where the fingers are with respect to the cube to touch it and then grasp it. However, since the LRN only computes relations to the goal, the fingers-cube relation is not explicitly available to the agent. The only way to manipulate the cube using available relations is to combine the cube-goal and fingers-goal relations using the goal as a reference. This capability means the agent can extrapolate and use higher-order relations. The LRN successfully solves the pushing task, which shows its ability to infer essential relations for the downstream task. The higher-order reasoning is handled by by the subsequent layers of the architecture after computing the relations, which is the MLP policy head in our case.

### D.3 Limitations of the Attention-Based Network

Given the observed poor generalization results and the very high variance during training for the attention-based baseline, we performed extensive hyperparameter tuning to ensure a fair comparison against our proposed RN and LRN modules. We used Bayesian search (Biewald, 2020) across more than 550 hyperparameter configurations. We found that none of those improves over our set of hyperparameters meaningfully.

To find potential bottlenecks in both the RL algorithm (PPO) and the attention modules, we used the hyperparameters search algorithm from Biewald (2020) to check elements of both. Regarding PPO, we tunned:

- Gradient clipping norm
- Discount factor
- Entropy loss coefficient
- Value loss coefficient
- GAE lambda
- Learning rate
- Ratio clip

For the archicture of the module we explored different configurations of:

- Embedding dimension
- Number of heads
- $\mathbf{f}_\phi$ output dimension

Due to the long training time, we evaluated each model after 50 million timesteps. This training length should be sufficiently long for the purpose of observing notable differences between models' fractional success and reward.

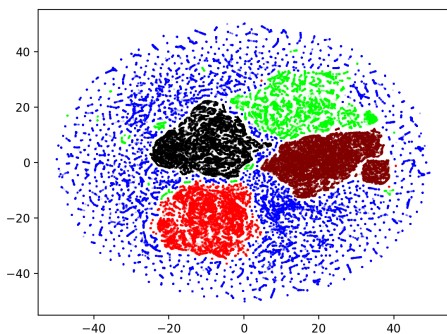

Figure 8: Low-dimensional t-SNE emebedding of the relations generated over 200 episodes from the Linear Relation Network. Different relations are color-coded (Black: Finger0-Goal. Brown: Finger1-Goal. Red: Finger2-Goal. Green: Cube-Goal. Blue: Distractors-Goal). The architecture learns to map different relations in different sub-spaces of the representation space.

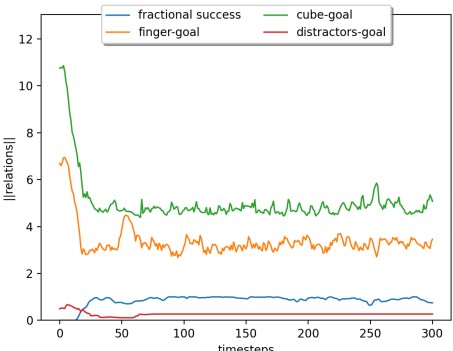

Figure 9: Relations norm changing throughout the task in LRN when testing on 4 distractors.

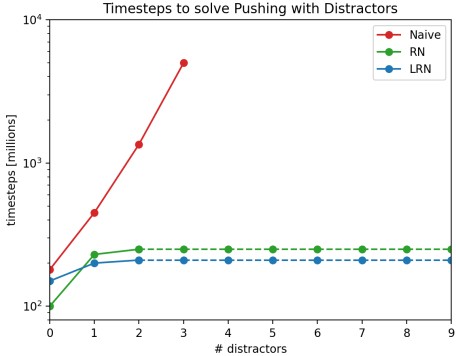

Figure 10: Scaling behaviour of training frames required solving to push a specified object to a target pose with additional distractors. Training agents with a naive fixed-input size MLP encoder is constrained to a predetermined object number and scales exponentially. Data points on dashed line are extrapolated from the our proposed models trained with only 2 distractors as they achieve zero-shot generalization for any more distractors.

