# OpenReview forum: "Compositional Multi-object Reinforcement Learning with Linear Relation Networks"
_ICLR.cc/2022/Workshop/OSC — ICLR2022 OSC  Poster_

### Official Review · Reviewer_Bycw · 2022-03-13

**Rating:** 2
**Confidence:** 3

**Review:**

**Summary**
In this work, the authors propose a linear relation network (LRN) to tackle manipulation tasks in zero-shot settings where an unseen number of objects are present. Specifically, it is assumed that only the number of distractor objects varies. The proposed LRN is a simple linearized version of the existing work, pairwise relation network. While the original relation network encodes the relation between each pair, the proposed LRN only models the relation between goal and each object, hence reducing the computation complexity from O(K^2) to O(K), where K is the number of objects. The proposed method was evaluated on the proposed multi-object manipulation domain. It was compared with relation network and attention model, and LRN outperformed the attention model by a large margin, and performed similarly to the relation network.


**Novelty**
Low. The proposed LRN is a simple and domain specific variation of the existing work RN.

**Relevance**
High. This work concerns modeling the relation between known objects, hence has high relevance to the workshop.


**Significance**
Low. The proposed model can only handle generalization over a number of distractor objects of the same kind, which is limited to only a specific setting.


**Soundness**
The paper is sound.


**Quality of writing/presentation**
The paper reads well.

**Comment**
The paper presents a reasonable model and the experiment result is at least positive for a specific task. However, it has relatively limited novelty and significance. It can be improved by more clearly presenting how the proposed model can be extended to a more general setting. Also, it would be great if the authors can provide more analysis on why ATTN model underperform the relational models.

---

### Official Review · Reviewer_asav · 2022-03-15
**Interesting Paper**

**Rating:** 2
**Confidence:** 3

**Review:**

The paper presents an architecture for reinforcement learning in the object manipulation setting. The main contribution is a network architecture called "Linear Relation Network," which is essentially a relation network (RN) that receives an additional goal object. Thus, the authors propose to consider only relationships with the goal object. Thus, the overall computation complexity is $O(n)$ instead of $O(n^2)$, where $n$ is the number of objects. The authors show that their model generalizes better to testing environments with more objects than those of training. The paper presents an interesting contribution to the literature and is relevant to the workshop.

My detailed weakness comments are the following.

1. What will happen if we just use a linear aggregation of object features instead of considering relationship between objects? Since there is only one goal object, I would imagine this method work works fine. See DeepSets as an example: https://arxiv.org/abs/1703.06114.
2. Assuming one single goal object is definitely limited. For example, even if we only care about pushing one object to the target region, there might be cases where we may actually need to push away other objects that are blocking our path. The authors should be more fair when comparing their approach with existing literature, for example Li et al "Towards practical multi-object manipulation using relational reinforcement learning." Exploring different design choices and see how they improves learning/generalization is important but should be done in a more systematical manner.
3. There have been a lot of work on reducing the quadratic computation complexity for relational architectures. For example, LinFormer https://arxiv.org/pdf/2006.04768.pdf.

---

### Decision · Program_Chairs · 2022-03-23

**Decision:**

Accept (Poster)

**Comment:**

The reviewers agree the paper should be accepted at the workshop. Congratulations!